# Periodontitis and Systemic Disorder—An Overview of Relation and Novel Treatment Modalities

**DOI:** 10.3390/pharmaceutics13081175

**Published:** 2021-07-30

**Authors:** Pooja Jain, Nazia Hassan, Karishma Khatoon, Mohd. Aamir Mirza, Punnoth Poonkuzhi Naseef, Mohamed Saheer Kuruniyan, Zeenat Iqbal

**Affiliations:** 1Department of Pharmaceutics, School of Pharmaceutical Education & Research, Jamia Hamdard, New Delhi 110062, India; pjain@jamiahamdard.ac.in (P.J.); naziahassan_sch@jamiahamdard.ac.in (N.H.); karishma3sep@gmail.com (K.K.); 2Department of Pharmaceutics, Moulana College of Pharmacy, Malappuram 679321, India; admin.P30@kuhs.ac.in; 3Department of Dental Technology, College of Applied Medical Sciences, King Khalid University, Abha 61421, Saudi Arabia; mkurunian@kku.edu.sa

**Keywords:** inflammation, cardiovascular diseases, diabetes mellitus, preterm birth, autoimmune disorders, cancer, periodonto-therapeutic

## Abstract

Periodontitis, a major oral disease, affects a vast majority of the population but has been often ignored without realizing its long-fetched effects on overall human health. A realization in recent years of its association with severe diseases such as carditis, low birth weight babies, and preeclampsia has instigated dedicated research in this area. In the arena of periodontal medicines, the studies of past decades suggest a link between human periodontal afflictions and certain systemic disorders such as cardiovascular diseases, diabetes mellitus, respiratory disorders, preterm birth, autoimmune disorders, and cancer. Although, the disease appears as a locoregional infection, the periodontal pathogens, in addition their metabolic products and systemic mediators, receive access to the bloodstream, thereby contributing to the development of systemic disorders. Mechanism-based insights into the disease pathogenesis and association are highly relevant and shall be useful in avoiding any systemic complications. This review presents an update of the mechanisms and relationships between chronic periodontal infection and systemic disorders. Attention is also given to highlighting the incidence in support of this relationship. In addition, an attempt is made to propose the various periodonto-therapeutic tools to apprise the readers about the availability of appropriate treatment for the disease at the earliest stage without allowing it to progress and cause systemic adverse effects.

## 1. Introduction

Periodontitis is one of the most prevalent dysbiotic oral diseases and is known to affect at least one tooth in 80% of adults worldwide [1]. A deep understanding of the etiology and pathogenesis of periodontal disease and its chronic, inflammatory, and infectious nature generates the need to recognize the possibility that it may have deleterious effects on other body parts as well [2,3].

Attention to this link was perhaps first recognized in the late 1980s when a group of scientists published data suggesting apposite observations such as the existence of periodontitis in addition to cardiovascular diseases was present in pregnant females facing premature labors and deliveries, as they suffered from periodontitis during the pregnancy [4]. Many studies have suggested the theory of focal infection in which an oral microorganism can migrate to adjacent or distant parts of the body. In the advanced stages of the disease, the pathogenic bacteria of dental plaque and their metabolic products may enter into the systemic circulation during mastication or mechanical procedures. Furthermore, the concept of periodontal medicine as a new discipline concentrating on validating this association and its biological mechanism through animal and humans has also been studied [5]. This review is an attempt to juxtapose the presence of periodontal infection in addition to systemic diseases such as cardiovascular disease, respiratory disorders, preeclampsia, glycemic control diseases, autoimmune disorders, and cancer. Additionally, a conscious effort is made to propose the various periodonto-therapeutic tools so as to apprise the reader about the availability of appropriate treatment for the disease at the earliest stage without allowing it to progress and cause systemic adverse effects.

## 2. Possible Mechanism behind the Systemic Manifestations of Periodontitis

As alluded to above, inflammation has a putative role to play in periodontitis. It is a programmed signaling episode aimed to protect the organism from infection. Infection acts as a stimulus for the release of pathogen associated molecular patterns (PAMPs) that then binds with the host cells’ receptors [6]. This binding initiates the inflammatory cascade which includes enhanced expression of inflammatory mediators and adhesion molecules. This further recruits macrophages, natural killer cells, dendritic cells, polymorphonuclear neutrophils (PMN), and other phagocytotic cells to the infection-affected tissue. Under normal circumstances, these phagocytic cells engulf and neutralize the infection, causing microorganisms which later undergo apoptosis on their own. Furthermore, removal of apoptotic cells signals a shift from pro to anti-inflammatory phenotypes, thereby curtailing the inflammation and restoring the tissues integrity [7]. Sometimes, however, even after the removal of pathogen, the inflammation cascade fails to switch off and this leads to uncontrolled and chronic inflammation (Figure 1). In advanced stages of periodontitis, when the immune cells are unable to control the propagation of the pathogenic bacteria, state of chronic inflammation may generate. Later on, this chronic inflammation serves as the hallmark for the bidirectional link between chronic periodontitis and systemic complications [8]. Moreover, this relation is consistently reported and highlighted by various researchers [6,9,10,11,12].

In general, the intact gingival epithelium of oral cavity acts as an innate physical barrier for the systemic spread of oral bacteria. In addition, the gingival epithelium is responsible for planning and maintaining the host inflammatory responses. During disease progression, the ulcerated epithelium loses its integrity and due to this, the connecting tissues and blood capillaries come in direct contact with the plaque biofilm. However, during mastication, the exposed ulcerated epithelium facilitates the invasion of bacteria into systemic circulation. Once the periodontal pathogens and their metabolic enzymes have access to the blood circulation, they produce inflammatory mediators which are responsible for systemic inflammation.

In a broad manner, the mechanism through which periodontal afflictions influence the systemic manifestation can be divided into 3 parts [13]:Oral-hematogenous migration of periodontal pathogens and its direct effects to target organs [14].Transtracheal migration of periodontal pathogens and its direct effects to target organs.Oral-hematogenous migration of inflammatory mediators such as cytokines and antibodies with their effects on distant organs.

## 3. Periodontitis and Systemic Complications

### 3.1. Periodontitis and Cardiovascular Diseases

A well-known relation exists between oral afflictions and cardiac disorders, where a bacterial load of oral infections is the major source of bacterial endocarditis and subsequent heart valve destruction as explained in Figure 2. Under the umbrella of CVD, atherosclerosis, coronary heart disease, endocarditis, and ischemic stroke are present and their association with periodontitis is well established by the two pioneer meta-analysis studies [15,16]. This relation is further explained below.

#### 3.1.1. Atherosclerosis

Atherosclerosis, a type of inflammatory disease, is believed to be the major cause of cardiovascular disease which is provoked by injury of the vascular endothelium. The disease progresses with plaque formation, plaque disruption, and subsequent atherothrombosis. However, the deposition of atheromatous plaques in the artery wall is indicative of atherosclerosis. Chronic infectious diseases can modify the type of inflammatory response in the artery wall as they directly supply pathogens into the bloodstream or somehow influence systemic inflammation [17]. Biological indicators such as C-reactive proteins (CRP), interleukin-6(IL-6), and tumor necrosis factor-α (TNF-α), which are expressed in cardiovascular disease, are the earliest sign of endothelial dysfunction due to systemic inflammation and have also been found to be increased in periodontitis-affected patients [18].

Periodontal bacteria enter the systemic circulation through a dental procedure such as tooth extraction, scaling, and periodontal probing. Sometimes, daily tooth care activity such as brushing may also lead to bacteremia and can induce low-grade inflammation mediated atherogenicity. However, the bacteria do not remain in circulation for a longer duration, but bacterial metabolic products such as endotoxins and gingipains trigger the systemic inflammatory responses. This initiates the production of inflammatory cytokines, the expression of adhesion molecules, and leukocyte infiltration on the cell surface, all of which contributes to atherosclerosis [19].

#### 3.1.2. Coronary Heart Disease

Coronary heart disease is the result of the deposition of thrombus or the atheromatous plaques inside the walls of coronary arteries. Several studies have shown the connection of infectious agents such as *P.gingivalis*, *Chlamydia pneumonia*, *Cytomegalovirus*, *Herpes simplex virus*, and *H. pylori* to atherosclerosis and coronary heart disease. A study conducted on an animal suggests that the *P.gingivalis* mediated bacteremia initiates coronary lesions and speeds up the coronary atherosclerosis [20]. Reports suggest that amongst patients with periodontitis, the risk of coronary heart disease increases up to 25%. Even tooth loss is a risk factor for an increased incidence of coronary heart disease and stroke [21].

#### 3.1.3. Stroke

The idea of a possible link between chronic inflammatory conditions and cardiac stroke has been mentioned in various studies and all the results indicate that chronic inflammatory infections such as periodontitis and cardiac stroke share common risk factors such as smoking habits, age, hypertension, and diabetes. In addition, elevated inflammatory markers of periodontitis are the indicators of cardiac stroke [22].

Until now, though, the relation between periodontitis and stroke was not revealed. Theories have been given to explain the mechanism of the relation between atherosclerotic plaque and stroke. Many of them focus on bacterial invasion which explains the direct effect of bacteria and their metabolites on the endothelium. The cytokine theory is based on the fact of host–parasite interaction due to which cells of the immune system release inflammatory mediators and these mediators have a key role in vascular endothelium wall damage [23]. Next is the theory of auto immunization that emphasizes the importance of heat shock proteins (HSP65) expressed on the periodontal pathogens such as *Porphyromonas gingivalis*, *Actinobacillus actinomycetemcomitans*, and *Prevotella intermedia* [24]. Bacterial lipopolysaccharides from the periodontal regions pass into the bloodstream and initiate the genesis of acute-phase proteins such as the C-reaction protein (CRP) in a patient of periodontitis-type chronic infection. Compared with the healthy controls, the CRP levels in periodontitis patients are consistently elevated. However, elevated CRP levels (~2.1 mg/L) are suggestive of a higher risk of stroke and acute thrombotic cardiovascular events [25]. It is believed that acute phase proteins are deposited in the damaged blood vessels and accelerate the phagocyte activation, and this nitrous oxide release promotes the atheroma formation during the process. Simultaneously, elevated CRP levels aggravate the inflammatory process at the atherosclerotic plaques in patients with periodontitis. In addition, the inflamed plaques are assumed to be unstable and prone to be ruptured, with increased chances of cardiovascular events [26].

### 3.2. Periodontitis and Autoimmune Disorders

In periodontitis, dysbiotic consortium of bacteria displaces the commensal microbiota of tooth surface that promotes chronic inflammation and destruction of tooth-supporting materials. In opposition to this rheumatoid arthritis is an auto-antibody triggered autoimmune disorder with a global prevalence of nearly 1%. Although the etiology of the two is different, both share similar pathology such as both involving pro-inflammatory cytokines that mediate chronic inflammation, connective tissue destruction, and bone erosion [27]. Several studies have been conducted to identify the relation between periodontitis and rheumatoid arthritis. It has been reported that patients with periodontitis have a higher prevalence of rheumatoid arthritis than patients without periodontitis [28].

It is hypothesized that oral infections play a crucial role in the pathogenesis of rheumatoid arthritis. The involvement of periodontal bacteria in the etiopathogenesis of rheumatoid arthritis is suggested by the successful treatment outcome of rheumatoid arthritis with antibiotics against anaerobic bacterial infections. It has also been observed that the synovial fluid of rheumatoid arthritis patients contains the DNA of anaerobic bacteria during later stages of periodontitis. The long-term presence of highly pathogenic bacteria in the oral cavity can develop chronic bacteremia that can damage even distant organs such as joints and endocardium [29,30].

It has been reported that the *P. gingivalis* is capable of altering the cell-epithelium integrity, penetrating endothelium cells, and disturbing protein synthesis. In this way, periodontal pathogens have a direct systemic entry into the blood circulation. In addition, blood and synovial fluid of rheumatoid arthritis patients show the DNA of periodontal bacteria and their specific antibody. Recently, it was observed that the *P. gingivalis* is capable of inducing cellular effects and penetrating the isolated primary human chondrocytes of knee joints. As a result of this penetration, the bacteria delayed the progression of the cell cycle and increased the chondrocytes’ cell apoptosis [31,32].

*P. gingivalis* produces enzymes that play an important role in bacterial housekeeping and infection. These enzymes are cysteine endopeptidase such as arginine-specific gingipain R and lysine-specific gingipain K. These enzymes are responsible for amino acid uptake from the host protein and maturation of bacterial fimbriae. As gingipains are proteolytic enzymes, they cause bacterial virulence by activating matrix metalloproteinases such as MMP-1, MMP-3, and MMP-95, and by degrading extracellular matrix host proteins such as collagen, fibronectin, and laminin. Gingipains also lead to increased vascular permeability and the degradation of complement factors [33,34].

In conclusion, a body of evidence built up over the past years unveils the relations of periodontitis with autoimmune disorders. In addition, periodontal pathogens can be considered as the direct trigger for autoimmune disorders.

### 3.3. Periodontitis and Respiratory Disorder

Respiratory disorders are known as the major cause of death and suffering of humans. There is evidence in the literature to support the fact that oral infections, particularly periodontitis, may influence respiratory disorders such as pneumonia and COPD (chronic obstructive pulmonary disease). The anatomical continuity between the lungs and oral cavity serves as the potential passage for respiratory pathogens. By aspiration of oropharyngeal secretions, oral pathogens may enter the lungs. However, in a healthy individual, the immunological and defense mechanism acts as a barrier for oral pathogens to reach the lower respiratory tract; to cause an infection, the pathogen must be virulent and the defense mechanism must be compromised [35].

The oral bacteria such as *Actinobacillus actinomycetemcomitans*, *P. gingivalis*, *Streptococcus constellatus*, *Prevotella intermedia*, *Capnocytophaga* species, *Actinomyces israelii*, and *Eikenella corrodens* are mainly responsible for lung abscesses and pneumonia [36].

To explain the role of oral bacterial pathogens in the etiology of respiratory disorders, the following mechanism has been proposed [35,36,37].

a. Aspiration of oral pathogens: Bacteria may influence the acceleration of COPD whereas dental plaque acts as a reservoir for the pathogenic bacteria that can cause lung abscesses or pneumonia. Reduced salivation and altered pH may promote respiratory pathogens’ colonization in ill or immune-compromised patients.

b. Mucosal surface modification: Periodontal bacteria secrets enzymes that may modify the mucous surface of the oropharyngeal cavity and promote the adhesion of respiratory pathogens. Modification of the mucous membrane may be due to the loss of fibronectin from the epithelial cell surface which exposes the mucous membrane receptors for bacterial cell adhesions.

c. Reduced clearance of the pathogenic bacteria: Patients with periodontal infections have increased levels of hydrolytic enzymes such as sialidase in their saliva. These enzymes alter the structure of mucins and destroy the salivary pellicle of bacteria, hence reducing its ability to bind and eliminate pathogens such as *H. influenza*.

d. Alteration of respiratory epithelium: Due to the host response, the cells of periodontium secrete a wide variety of cytokines and other biologically active factors such as interleukin (IL)-1α, IL-1β, IL-6, IL-8 and TNF-α. Cytokines from the periodontal tissue pass from the gingival sulcus, are mixed with the saliva, and may stimulate the respiratory epithelial cells. These stimulated epithelial cells then release other cytokines and attract inflammatory cells to this site. Furthermore, these inflammatory cells secrete hydrolytic enzymes that can damage the epithelium and the damaged epithelium is more prone to colonization by respiratory pathogens. By reviewing the pathogenic relations and epidemiological studies, it seems that early diagnosis and treatment of periodontal disease might reduce the overt events of chronic obstructive diseases and good oral health is of prime importance for the prevention and control of respiratory disorders [38].

### 3.4. Periodontitis and Diabetes Mellitus

Periodontitis has various damaging effects on quality of life. Epidemiological data indicated diabetes as a major risk factor for periodontitis and chances to develop periodontitis is increased by approximately three times in a diabetic patient. There is also a clear relationship between the extent of hyperglycemia and the severity of periodontitis. There are emerging pieces of evidence in favor of the two-way relationship between diabetes and periodontitis, In addition, periodontitis treatment is associated with an approximately 0.4% reduction in HbA1c [39]. Furthermore, the two-way relationship is explained in the following section.

#### 3.4.1. Periodontitis as a Consequence of Diabetes

It has been observed that in both types of patients of diabetes such as diabetes mellitus type 1 and type 2, high levels of systemic inflammatory biomarkers are present. The result of elevated inflammatory conditions leads to microvascular and macrovascular complications and this hyperglycemia may potentiate the genesis of oxidative stress, cell apoptosis, and inflammation. In diabetes and obesity, increased levels of IL-6 and TNF-α have been demonstrated, whereas the serum levels of IL-6 and CRP have been shown to predict the future occurrence of diabetes mellitus type 2. An elevated CRP level is the marker of diabetes mellitus, insulin resistance, and cardiovascular disease. IL-6 and TNF-α are important inducers of acute-phase proteins such as CRP and have the capacity to impair intracellular insulin signaling, predominantly contributing to insulin resistance. Periodontitis is also associated with elevated levels of CRP and IL-6 with varying levels of IL-6 accompanied with disease progression [40]. Hence, the periodontitis-associated systemic inflammation aggravates the diabetic state.

The deposition of advanced glycation end-products (AGEs) in periodontal tissue also plays a key role in the up-regulation of periodontal inflammation in diabetic patients. AGE and its receptor binding led to the increased production of inflammatory mediators including IL-6, TNF-α, and IL-1β-. AGE formation leads to the production of reactive oxygen species (ROS) that promote oxidant stress and further endothelial cell damage with a vascular injury which is a complication of diabetes. AGEs also increase the respiratory burst in polymorphonuclear neutrophils (PMNs) which can increase the local tissue damage in periodontitis. AGEs have deleterious effects on bone metabolism as well; it impairs bone formation and decreases the production of the extracellular matrix [41].

Studies based on an osteoclastogenic factor have reported elevated levels of the receptor activator of nuclear factor kappa-Β ligand (RANKL) in diabetes-associated periodontal tissue. It has been proposed that hyperglycemia may alter the RANKL/OPG (osteoprotegerin) ratio in periodontal tissue and this might indicate the speedy alveolar bone destruction in diabetes [42].

#### 3.4.2. Diabetes as a Complication of Periodontitis

An intriguing area of recent investigation has focused on whether periodontitis plays a role in the incidence of diabetes. A dysregulated immune system is central to the pathogenesis of diabetes and its associated complications. Type 2 diabetes (T2DM) and related conditions such as obesity are associated with some physiological, nutritional, and metabolic changes including hyperglycemia, production of advanced glycation end-products (AGEs), hyperlipidemia, and increased adiposity; these changes have several consequences including immune-dysregulation as manifested by a pronounced, long-lasting inflammatory state and weakened self-limitation and resolution of immune responses [12,43]. Figure 3 explains the two-way relation between periodontitis and diabetes.

Although there is substantial evidence to support the fact that periodontitis is a complication of diabetes, simultaneously, there is also clinical evidence to support the inverse, i.e., periodontal infection adversely affects glycemic control in diabetics. Highly vascular inflamed periodontium may act as a reservoir or as endocrines like source of TNF-α and other mediators of inflammation. In addition, the predominance of gram-negative periodontal pathogens in the ulcerated pocket epithelium serves as a chronic source of bacterial products and locally produced inflammatory mediators. It has been shown that particularly after an acute infectious trauma, the important mediators of periodontal inflammation such as TNF-α, IL-6, and IL-1 alter the glucose and lipid metabolism. Evidence suggests that TNF-α interferes with lipid metabolism and acts as an antagonist of insulin. It has also been reported that IL-6 and IL-1 antagonize insulin action. Surprisingly, all reports of altered endocrinology functions suggest the presence of acute infectious conditions [44,45].

### 3.5. Periodontitis and Pre-Term Low Birth Weight Babies

Preterm infants are born before the completion of the 37-week gestation period. Around 11% of pregnancies end in preterm birth and are the most important cause of infant mortality. Despite the advancements in post-natal care, infant mortality is rising and still there is a need to understand the mechanism behind adverse pregnancy-related outcomes [46].

It has been proven that maternal intrauterine infections and bacterial vaginosis play an important role in pregnancy-related complications such as preterm birth. A clear-cut mechanism to explain the relations is still not available in the literature but it is believed that the humoral inflammatory responses of the mother and fetus are responsible for preterm birth. Infection-mediated genetic variation is also one of the possible risk factors for preterm birth [47].

Epidemiological, microbiological, and clinical evidence exists to support the association between maternal infection and preterm birth. As revealed by the epidemiological studies of spontaneous preterm birth, a gestation period of fewer than 34 weeks is much more frequently associated with clinical or subclinical infection than those at more than 34 weeks. In addition, the potency of a link between clinical and subclinical infections increases as gestational age decreases, especially before 30–32 weeks. The risk of both maternal and neonatal infections increases as the gestational age decreases and both infections are more common after a preterm birth than a term birth. Infections are not only the contributor of significant preterm births but they also strongly contribute to those preterm births that are responsible for significant infant morbidity and mortality. Maternal genitourinary and reproductive tract infections have been implicated as the main risk factor in 15–25% of preterm deliveries [48].

The existence of a relationship between maternal periodontal infection and delivery of a preterm low birth weight baby has become a trending topic as the gram-negative anaerobic bacteria of a periodontal cavity may serve as a reservoir of endotoxin and lipopolysaccharides that then lead to an increase of local inflammatory mediators such as PGE_2_ and cytokines, and ultimately these systemic inflammatory mediators lead to preterm birth [49]. In many observational studies, the humoral responses towards the oral pathogens of the mother and fetus showed that there was a 2.9-times higher incidence rate of fetal IgM seropositivity for oral pathogens among preterm babies as compared with term babies. The absence of the maternal IgG antibody to oral pathogens was linked with an increased frequency of preterm birth. The highest frequency of preterm birth was found among those mothers with no IgG response to oral pathogens who delivered an infant that showed an IgM response. Thus, it could be stated that maternal periodontitis without a defensive antibody response is linked with the systemic translocation of an oral pathogen to the fetus that results in preterm delivery [50], and they hypothesized the archetype as presented in Figure 4.

Altogether, these studies prove that periodontal infections are a major risk factor for preterm low birth weight babies.

### 3.6. Periodontitis and Cancer

Cancer, characterized as a heterogeneous group of disorders having different biological properties, is supposed to be arising from a single cell in which genetic alterations in tumor-suppressor genes and oncogenes are responsible for the continued clonal selection and tumor cell heterogeneity, resulting in tumor proliferation, metastasis, invasion, and drug resistance. Cancer-associated inflammation is considered the seventh sign of cancer. Although inflammation can play a role to suppress a tumor by stimulating an antitumor immune response, under certain conditions, it stimulates tumor development. This apparent contradiction can be explained by the nature and intensity of inflammation [52].

It has been observed that cancer-associated inflammation is similar to chronic inflammation. The association between inflammation and cancer can be supposed of consisting of two mechanisms: an extrinsic pathway, in which a constant inflammatory state leads to increased cancer risk, and an intrinsic pathway, in which acquired genetic variations trigger a tumor development. The relationship between the two pathways of the cancer development process relies on their specific interactions between genetic/epigenetic factors and environmental factors. Several epidemiological studies indicated a link between periodontal disease and cancer in different tissues [53].

The mechanism of periodontitis-induced carcinogenesis could also differ by site. For instance, bacteria may have a more direct role in carcinogenesis in the mouth or lung, whereas in more distant organs, systemic inflammation, or nitrosamines, reactive oxygen species (ROS) may play a more important role. Various possible mechanisms linking periodontitis and cancer includes the following [54,55]:Periodontal pathogens induce chronic inflammation; this promotes already initiated cells, leading to uncontrolled cell growth and potential carcinogenesis.Periodontopathic bacteria may also have a more direct role through local inflammatory responses and carcinogenic transformations. *Helicobacter pylori* infection is an example of this.Chronic periodontal disease may suggest that an individual’s immune system is compromised, unable to clear the infection, and consequently deficient at surveillance for tumor growth.Periodontal inflammation can lead to genetic alteration via the production of reactive oxygen and nitrogen species-type oxidizing compounds.Carcinogenesis and wound healing shared several common biological processes and carcinogenesis can be considered as an unregulated form of wound healing.Failure of the periodontal inflammation-resolving mechanism.

## 4. Current Treatment Modalities and Advanced Pharmaceutical Approach

Although the clinical mainstay treatment of periodontitis revolves around the use of various types of antibiotics in addition to other symptomatic relief drugs such as anti-inflammatory drugs and others, the continuous efforts of pharmaceutical researchers have led to the design and development of certain advanced dento-therapeutics which are primarily focused on spatio–temporal drug delivery for a prolonged period [56]. A recce of concerned literature has revealed the prevalence of various formulation designs focusing on locoregional effects. Table 1 summarizes the various pharmaceutical research designs with a primary objective of the formulation.

The conventional treatment modalities for periodontal infections primarily focus on dental plaque, which is an oral biofilm responsible for causing various inflammatory reactions in periodontal tissues. Different available approaches to target dental plaque include plaque control, non-surgical, and surgical interventions. Adjuvant therapies such as antibiotics or supplements are also included, although they face major limitations of antibiotic resistance, local inflammation in the periodontium, and host immune responses, thus causing partial effectiveness [57]. Therefore, advanced strategies to mitigate, treat, and lessen periodontal infections have become a need of the present time. Recently, various modalities such as quorum sensing inhibitors, inflammasome targeting, host inflammatory substances, bone immune responses, and FDA-approved anabolic agents, namely, the teriparatide and sclerostin antibody, have been introduced as a step forward against periodontal infections [58]. Quorum-sensing inhibitors target molecules that modulate microbial signaling mechanisms, a primary step in biofilm formation, thus inhibiting plaque biofilm formation and controlling periodontal infections [59]. The inflammasome is a naturally produced cytosolic multiprotein oligomer of the innate immune system. Abnormal inflammasome activation is associated with various ailments including periodontal disease pathogenesis. Thus, the development of drugs that directly target and inhibit an inflammasome activation may hinder the progression of the periodontal infection [60]. The FDA-approved anabolic agents for bone regeneration could be incorporated with current treatment regimens as dental implants for severe cases of periodontitis [58]. Reportedly, the use of photodynamic therapy has also gained considerable attention against dental plaque; for example, the potential use of nano-based antibacterial photodynamic therapies to combat bacterial plaque-initiated oral diseases and adjunctive application of antimicrobial photodynamic therapy in nonsurgical periodontal treatment has been noted [61]. Various developing strategies such as bioactive-based dental polymers, nano-size building blocks, and bio-implants are also being considered as new generation dental restoration tools and can inhibit oral biofilms [57].

## 5. Conclusions

Periodontitis is no longer a standalone disease and transcends its effect beyond the site of its infection in the oral cavity. The unravelling of the underpinnings of its association with a plethora of systemic diseases have put both the medical practitioners as well as researchers on alert. There seems to be an unmet need of collating scientifically derived data in the aforementioned domain so as to support the therapeutic decision-making of physicians. As a part of therapeutic strategy, it is advisable to conduct a discernible assessment of the periodontal infections while diagnosing apparently unrelated diseases such as atherosclerosis, diabetes, and even cancer. There has been a remarkable progression in devising various novel dental therapeutics, adding both volume and quality to the dentists’ armamentarium which if used judiciously would lead to the faster and earliest amelioration of these site-specific diseases. The novel pharmaceutical interventions are majorly laden with attributes of reduced drug doses and sustained effects which make them economically viable. Additionally, an early recognition and treatment of the disease shall be a great stepping stone towards limiting or precluding its adverse effect(s) on other systemic disorders and would substantially improve patient outcomes.

## Figures and Tables

**Figure 1 pharmaceutics-13-01175-f001:**
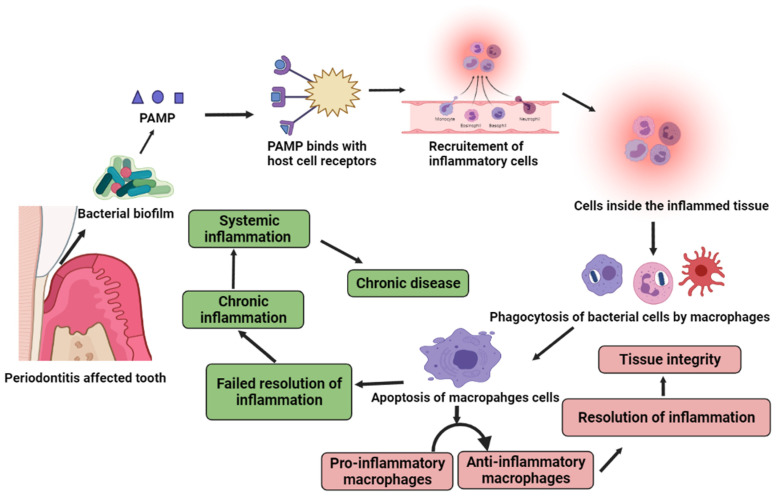
Inflammatory cascade of periodontitis. Pathogen mediates a series of events leading to the recruitment of inflammatory cells at the infection site. This triggers the phagocytosis of bacterial cells, followed by the apoptosis of phagocytic cells and resolution of inflammation. Imbalance in this cascade leads to chronic inflammation and associated systemic complications.

**Figure 2 pharmaceutics-13-01175-f002:**
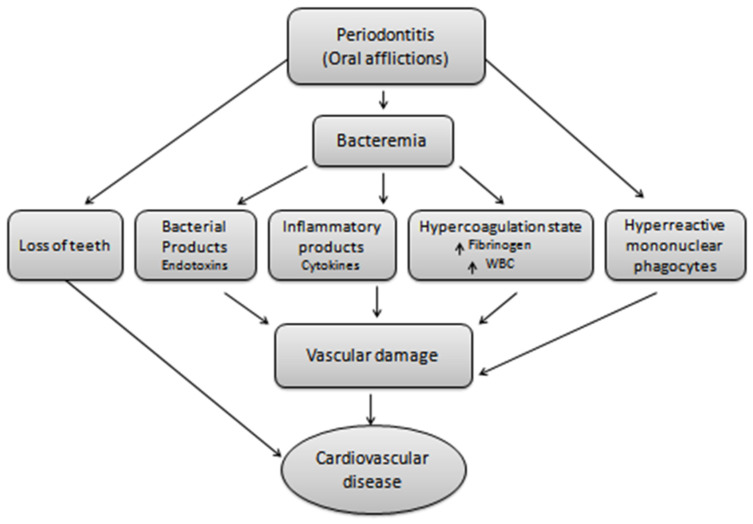
Oral afflictions and cardiovascular disease. Oral infections mediated bacteremia generates the hyper-reactive mononuclear phagocytes, bacterial and inflammatory products, a state of hyper-coagulation and teeth loss. All this leads to vascular damage and hence results in cardiovascular complications.

**Figure 3 pharmaceutics-13-01175-f003:**
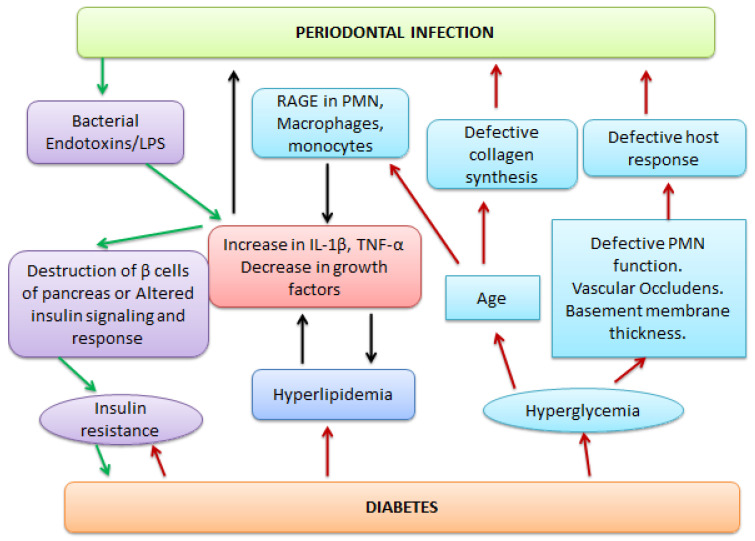
Two-way link of periodontitis with diabetes. Bacterial endotoxins lead to an increase in interleukin (IL-1β) and the tissue necrosis factor (TNF-α) which results in hyperlipidemia. This also leads to the destruction of β cells of the pancreas and hence insulin resistance. In the reverse direction, the hyperglycemia leads to defective polymorphonuclear neutrophils’ (PMN) functions, defective host response, and collagen synthesis, which promotes RAGE (receptors of advanced glycation end-products) in PMN. All these factors ultimately aggravate the periodontal infection.

**Figure 4 pharmaceutics-13-01175-f004:**
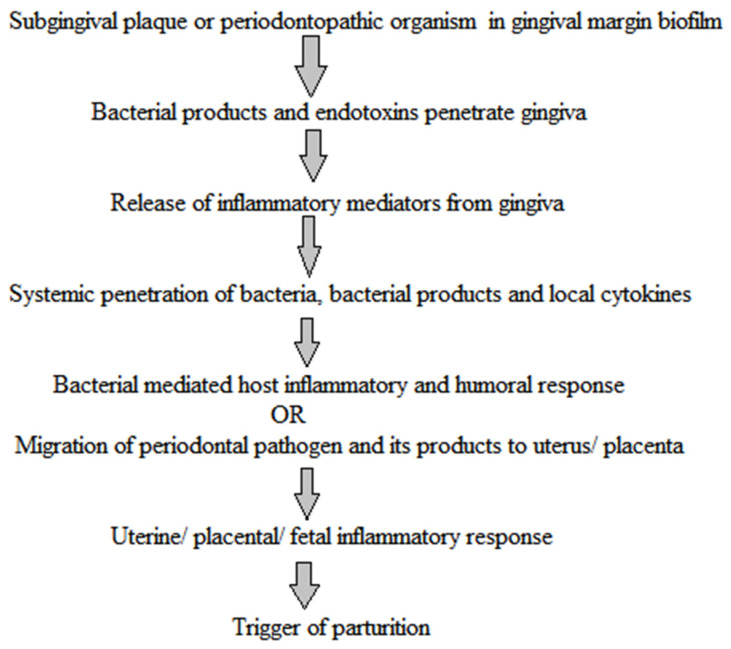
Hypothetical translocation of periodontal infection to fetus. Periodontopathic bacteria and their endotoxins penetrate the placenta from the gingiva via systemic circulation. This generates the inflammatory response in the placenta, resulting in the trigger of parturition [51].

**Table 1 pharmaceutics-13-01175-t001:** A summary of periodontal formulations and their applications.

Formulation	Components/ Devices	Intervention/ Condition	Description	Observations	Reference
Nanofibrous scaffolds	Silver nanoparticles, AgNPs and hydroxyapatite nanoparticles, and HANPs electrospun to prepare nanofibrous composites based on polylactic acid/cellulose acetate (PLA/CA) or polycaprolactone (PCL) polymers	Periodontal tissue and bone regeneration	Biodegradable electrospun nanoparticles-in-nanofibers-based scaffolds for guided tissue regeneration (GTR) and guided bone regeneration (GBR) with enhanced mechanical properties, cell adhesion, biocompatibility, and antibacterial properties	In-vitro studies of nanofibrous films cut into 10 × 10 mm^2^ samples showed that the addition of HANPs improved the cell viability by about 50 and AgNPs provided sustained antibacterial activity (40 mm zone of inhibition diameter) for 32 days. Additionally, the nanofibrous scaffolds offer optimum mechanical properties, tensile modulus (20–38 MPa), and a desirable degradation profile (40–70% of its mass in eight weeks).	[62]
Mineralized nanofiber	PLGA–collagen–gelatin coupled with calcium-binding bone morphogenetic protein 2 (BMP-2)	Alveolar bone regeneration	Mineralized nanofiber segments (20 μm) coupled with BMP-2, mimicking peptides for periodontal bone regeneration	In animal studies, the mineralized nanofibers were implanted into critical-sized maxillary defects of 2 mm in diameter and 2 mm in depth in rats and a sustained release profile was recorded for over four weeks. X-ray microcomputed tomography (μ-CT) analysis revealed ~3 times greater new bone volume and mineral density.	[63]
Polymeric fibers	Electrospun nanofibers encapsulation BAR using poly (lactic-co-glycolic acid) (PLGA), poly (L-lactic acid) (PLLA), and polycaprolactone (PCL) either as a single or blended solution with polyethylene oxide (PEO)	Inhibition of *Porphyromonas gingivalis* and adherence to *Streptococcus gordonii*	Rapid-release polymeric electrospun nanofibers against *P. gingivalis*/*S. gordonii* biofilms in-vitro	The most promising formulation of 10:90 PLGA: PEO of electrospun nanofibers has demonstrated a 95% BAR release after 4 h, a dose-dependent inhibition of biofilm formation (IC50 = 1.3 μM), disruption of established dual-species biofilms (IC50 = 2 μM), and maintenance of high-cell viability.	[64]
Nanofibrous membrane	PLLA/gelatin	Periodontal tissue regeneration	Biodegradable multifunctional nanofibrous membrane prepared by electrospinning biodegradable polymers with magnesium oxide nanoparticles (nMgO) for periodontal tissue regeneration and high antibacterial capacity	In-vitro results showed that incorporating nMgO into poly (lactic acid) (PLA)/gelatin elevates the tensile strength to maintain structural stability and adjust the degradation rate for periodontal regeneration. Considerable antibacterial and osteogenic properties were also observed. The in-vivo investigations in a rat periodontal defect model demonstrated effective periodontal tissue regeneration guided via nMgO-incorporated membranes.	[65]
Polymeric films	PLLA/PCL blends containing propolis	Guided periodontal tissue regeneration	Biodegradable composite membranes produced from PCL/PLLA blends with a natural antibacterial extract (propolis) as novel periodontal barrier membrane	The in-vitro antibacterial studies revealed remarkable activities against *Staphylococcus aureus* (17 mm zone of inhibition). The prepared films also showed faster degradation in physiological conditions.	[66]
Biopolymer composite film	Curcumin	Topical patches for wound care, periodontitis, and oral cancer treatment	Multifunctional biopolymer composites based on curcumin-loaded bacterial cellulose/alginate/gelatin	The in-vitro studies have shown substantial antibacterial activity against *E. coli* and *S. aureus* infection. The purported composite films exhibited cytotoxicity to human keratinocytes and human gingival fibroblasts, and also show potent anticancer activity in oral cancer cells.	[67]
Regenerative scaffolds	Cellulose hydrogels and biopolymers derived from plants, and Larrea tridentate	Periodontal tissue regeneration	Cellulose hydrogel films enriched with LT for biomedical application in wound healing and as regenerative scaffolds	For in-vitro studies, NIH3T3 mouse embryonic cells were used for the measurements of cell viability and morphology assays. For in-vivo assays, hydrogel films were implanted intramuscularly into female Wistar rats (250 g weight; 2 months), to analyze their cytocompatibility and biocompatibility.	[68]
Nanofibrous membrane	PLGA/gelatin, dexamethasone (osteogenic), and doxycycline hyclate (anti-bacterial agent)	Guided bone regeneration	Bi-layered electrospun composite nano-membrane with combined osteogenic and antibacterial properties for guided bone regeneration	In-vitro studies indicated that both dexamethasone and doxycycline hyclate followed a favorable sustained drug release profile. The cell viability evaluation suggested good cytocompatibility. The osteogenesis analyses demonstrated an enhanced osteoinductive capacity for rat bone marrow stem cells, increased alkaline phosphatase activity, enhanced calcium deposition, and upregulated osteocalcin expression. Furthermore, the antimicrobial experiments revealed effective antibacterial potency.	[69,70]
Hydrogel	Polyacrylic acid (PAA) hydrogel containing metronidazole	Therapeutic dressing	Gamma-ray irradiation targeted metronidazole-loaded PAA hydrogel	The in-vitro cytocompatibility test was performed according to ISO 10993-5 and the formulation exhibited no cytotoxicity. The antibacterial activity against *E. coli* (ATCC 43895), *S. aureus* (ATCC 14458), and *S. mutans* (ATCC 25175) yielded satisfactory results. In release studies, metronidazole from the PAA hydrogel was consistently released and reached approximately 80% at 120 min.	[71]
Hydrogel	Doxycycline/ lipoxin and poly isocyano peptide (PIC)	Periodontal	Antimicrobial and anti-inflammatory thermo-reversible hydrogel for improved gingival clinical attachment and periodontal drug delivery	The formulations were characterized in-vitro and in dogs with naturally occurring periodontitis. The results showed that the prepared hydrogel could be easily injected into periodontal pockets due to the thermo-reversible nature of the material. The formulation yielded significant release with no local or systemic adverse effects. A reduced subgingival bacterial load, pro-inflammatory interleukin-8 level, and improved gingival clinical attachment by 0.6 mm was also observed.	[72]
Liposomal gel	Lidocaine/ prilocaine	Periodontal	A randomized, double-blind, cross-over, and placebo-controlled clinical trial of liposomal gel (intra pocket) for non-invasive anesthesia in scaling and root during periodontal therapy	The sample size calculation was based on pain intensity (primary outcome) using visual analogue scale (VAS) data. The study reported no difference between intervention groups concerning pain frequency/intensity (primary outcome). The anesthetic gel did not interfere with the hemodynamic parameters (secondary outcome). However, the above observations have a few limitations: First, there is no ideal scale for measuring pain and hence further clarification is necessary. Second, periodontal procedures usually cause low or moderate pain. Third, there was low patient compliance as many did not prefer local anesthesia.	[73]
Gel	Doxycycline encapsulated in β-cyclodextrin	Periodontitis	A randomized, blinded clinical trial to compare the effects of 10% doxycycline gel with doxycycline encapsulated in β-cyclodextrin gel on 33 subjects with periodontitis for 30 days.	The adjunctive topical agents (doxycycline encapsulated in β-cyclodextrin gel) along with scaling and root planning resulted in significant improvements in clinical periodontal parameters such as visible plaque index, measurement of periodontal probing depth, clinical attachment level, and bleeding on probing.	[74]
Hybrid hydrogels	Mesoporous silica, minocycline, silver, and gelatin methacrylate	Periodontal infection	Near-infrared light (NIR)-activated hybrid hydrogels	The hybrid hydrogels showed controllable minocycline delivery with increased release rates (in-vitro). The hydrogels also exhibited synergistic antibacterial activity (90%) against *Porphyromonas gingivalis*. The photothermal treatment was as high as 66.7% against *P. gingivalis* as well to rapidly eliminate and maintain low bacterial retention in the periodontal pockets. Furthermore, the in-vitro cytotoxicity studies revealed an 80% cell viability.	[75]
Hydrogel nanoparticles	Minocycline, zinc oxide, and serum albumin	Periodontitis	Broad-spectrum hydrogel-based minocycline and zinc oxide-loaded serum albumin nanoparticles for periodontitis application with low toxicity and high antimicrobial and antibacterial activity	In in-vitro analysis, a slow-release time was observed. Encapsulation efficiency was 99.99%. The in-vitro skin adhesion experiment showed a bioadhesive force of 0.35 N. Broad-spectrum antimicrobial and antibacterial ability and high cell survival rates with low toxicity were observed.	[76]
Microspheres	PLGA, PIC, doxycycline, and lipoxin	Periodontal infection	A tunable and injectable localized system based on PLGA microspheres, containing doxycycline and lipoxin, dispersed into thermo-reversible PIC hydrogel for personalized periodontal application	The in-vitro efficacy and bioactivity of the released doxycycline had presented a comparable zone of inhibition with respect to fresh or unbound drugs against gram-negative anaerobic bacteria Porphyromonas gingivalis (ATCC 33277). The fluorescent bead internalization assay of lipoxin revealed that more fluorescent beads were internalized that may stimulate RAW264.7 macrophage (Gibco) phagocytosis. The in-vivo test on ten 8-week-old male Wistar rats (~250 g) had exhibited no obvious inflammatory responses.	[77]
Combination gel	Chlorhexidine and metronidazole	Gingivitis	A triple-blind randomized clinical trial on 90 subjects to compare and assess 0.8% metronidazole gel, 0.2% chlorhexidine gel, and the alternate application of the two gels against dental plaque and gingivitis for 14 days	The primary outcome measures are the bleeding index. The secondary outcome measures are the oral hygiene index, probing depth, and gingival index. The aforementioned outcomes were compared after 2 and 6 weeks.	[78]
Chitosan templates	Chitosan, glutaraldehyde, and doxycycline hyclate	Periodontal tissue regeneration	Cross-linking doxycycline-loaded freeze gelated chitosan templates for periodontitis	The in-vitro analysis of chitosan templates through a conventional dialysis sac method showed a 40 μg/mL of release after 24 h. Such a suitable drug release rate will also limit the toxicological effect of the cross-linking agent.	[79]
Retraction gels	Epinephrine, tetrahydrozoline, oxymetazoline, and phenylephrine	Gingival retraction	In-vitro vaso-constrictive retraction agents against primary human gingival fibroblasts in periodontal tissues	Immunocytochemical analysis revealed the biological effect of retraction gels on the expression of collagen types I and III. The generation of the reactive oxygen species triggered by the retraction gels indicated oxidative stress similar to the control cells using the dichlorofluorescein (DCF) fluorescent probe.	[80]
In-situ gel	Doxycycline hyclate, shellac, ethocel, and eudragit RS	Periodontitis	In-situ forming gels for localized periodontal pocket delivery.	The in-vitro release study through a dialysis membrane follows a sustained release pattern. It also exhibited in-vitro degradability and the antimicrobial effect against *S. aureus*, *S mutans*, *E. coli*, *P. gingivalis*, and *C. albicans.*	[81]
Nanofiber based hydrogel	Cellulose, κ-carrageenan oligosaccharide, surfactin, and herbmedotcin	Periodontitis	Anti-microbial loaded cellulose nanofiber and κ-carrageenan oligosaccharide composite hydrogels for strong antibacterial activity against periodontal pathogens such as *Streptococcus mutans*, *Porphyromonas gingivalis*, *Fusobacterium nucleatum*, and *Pseudomonas aeruginosa* in periodontitis treatment	Purportedly, they reduce the reactive oxygen species (ROS) generation, transcription factor, and cytokine production in human gingival fibroblast cells (HGF) under inflammatory conditions.	[82]
Hydrosilver gel	Silver	Chronic periodontitis	A prospective longitudinal pilot study using polymerase chain reaction analysis of hydrosilver gel against dental plaque in chronic periodontitis	The in-vivo model of chronic periodontitis was used for 15 days. The LAB^®^-test (LAB s.r.l.^®^, Ferrara, Italy) detected and quantified the presence and level of the most involved periodontitis pathogens that constitute the ‘red complex’: *P. gingivalis, Tannerella forsythia* and *Treponema denticola*. Other bacteria of the ‘orange complex’ were also monitored, such as *Fusobacterium nucleatum*, *Campylobacter rectus*, *Aggregatibacter actinomycetemcomitans*, *Atopobium rimae*, *Eubacterium saphenum*, *Porphyromonas endodontalis*, and *Treponema lecithinolyticum*, as the main components of microbiological shift.	[83]
Microporous annealed particle (MAP) hydrogels	Poly(ethylene) glycol	Tissue engineering and regeneration	Versatile new platform for the delivery of human periodontal ligament stem cells and periodontal tissue regeneration	In-vitro characterization revealed excellent retention, proliferation, and spreading of platelet-derived growth factors and human periodontal ligament stem cells within hydrogels.	[84]
Exosomal nanoparticles	Ginger phosphatidic acid	Oral biofilms	Plant-derived nanoparticles to inhibit *P. gingivalis* biofilm	Demonstrated inhibition of *P. gingivalis* induced bone loss and pathogenicity in an in-vivo mouse model of chronic periodontitis	[85]
Mesoporous nanospheres	Ipriflavone	Periodontal infection	Ipriflavone-loaded mesoporous nanospheres for periodontal augmentation	Periodontal augmentation was observed in an in-vitro osteogenesis model (MC3T3-E1 osteoprogenitor cells).	[86]
Nanocomposites	Chlorin e6	Periodontal diseases	A photodynamic therapy-guided bioactive nanocomposite containing chlorin e6 as a photosensitizer against biofilms on dentin squares. The dentin samples were prepared from extracted caries-free human molars that serve as the substrates for biofilm formation.	Photosensitizer effect on *Porphyromonas gingivalis*, *Prevotella intermedia*, and *Fusobacterium nucleatum* and their corresponding biofilms on dentin squares 5 × 5 mm (thickness of about 1 mm).	[87]
Nanocomplexes	Bovine serum albumin	Periodontitis	Nanocomplexes for enhanced osteogenic differentiation of inflammatory periodontal ligament stem cells	The in-vitro hemolysis assay and in-vivo cytocompatibility assay using BALB/c mice (8 weeks old) revealed the high transfection efficiency and biocompatibility of the prepared nano complexes.	[88]
Carbon quantum dots	Tinidazole and metronidazole	Oral biofilms	Periodontitis treatment by penetrating the *P. gingivalis* biofilm and destroying its related genes	An in-vitro biofilm penetration assay revealed that nanoscale tinidazole carbon quantum dots can penetrate through the biofilm to induce significant inhibition of *P. gingivalis* under the biofilm. In addition, as exhibited in the in-vitro antibacterial assay, tinidazole carbon quantum dots impair toxicity and inhibit the major virulence factors and related genes involved in the biofilm formation of *P. gingivalis*.	[89]
Nanoplatelets	Fluoride	Periodontal bone tissue regeneration	Osteogenic differentiation of human dental follicle stem cells for tissue regeneration	MTS assay and cellular morphology analysis demonstrated low cytotoxicity of prepared nanoplatelets at low concentrations.	[90]

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
