# Peer review of "Periodontitis and Systemic Disorder—An Overview of Relation and Novel Treatment Modalities"

_pharmaceutics, 2021, doi:10.3390/pharmaceutics13081175_

Round 1

Reviewer 1 Report

TITLE:  Periodontitis and Systemic Disorders- An Overview of Relation and Novel Treatment Modalities

Pharmaceutics MDPI

The aim of the present investigation was to provide a literature review on Periodontitis and Systemic Disorders.

GENERAL COMMENTS

The article is not completely in line with the journal topic that according to the Journal scope should imclude : “a provide concise and precise updates on the latest progress made in a given area of research. Systematic reviews should follow the PRISMA guidelines.” A wide part of this paper is oriented to a incomplete description of the relationship between the periodontitis and many different systemic diseases.

The introductive descriptive part should investigate deeply the cellular response mechanisms and the signal cascade that supports the relationship between the periodontal disease and the systemic diseases citing more clinical trials and histological studies. This paper did introduce no apparent organic methodologies for the papers selection. Moreover, the 3rd paragraph about the current treatment modalities and advanced pharmaceutical approach included a too heterogeneous studies with very few information about the study design (RCT, in vitro studies?), human/animal model, sample size, administration protocol and dose/effect. The discussion section is absent, and the conclusion appeared a little bit inconsistent.

Other comments.

The fig. 2 is too low in quality and should be improved.

I suggest to expand the discussion section after reading the follow paper: 33804123

The reference format should be adapted according the journal guidelines.

Reviewer 2 Report

This manuscript is an interesting review describing the Relation and novel treatment modalities between periodontitis and systemic disorders. The review is fundamentally interesting and could potentially provide understanding the complications between periodontitis and systemic diseases. However, there are some minor mistakes.

Line 168

“the lower respiratory 26tract”   Please omit 26.

Line 170 - 172

Bacterial species should be italic.

Line 275

Why does the tile is bold? It may be included in 2.5 section.

Line 308

figure 3 to Figure 3

Table 1

The right side of the table is not completed.
